# Causes of admissions and in-hospital mortality among patients admitted to critical care units in primary and secondary hospitals in Vietnam in 2018: a multicentre retrospective study

Vu Quoc Dat  ,[1] Bui Thi Khanh Linh  ,[1] Giang Bao Kim[2]

[1]Department of Infectious Diseases, Hanoi Medical University, Hanoi, Viet Nam
[2]Institute for Preventive Medicine and Public Health, Hanoi Medical University, Hanoi, Viet Nam

**Correspondence to**
Dr Vu Quoc Dat;
quocdat181@yahoo.com

## ABSTRACT

**Objective** The goal of this study was to describe the burden of disease and in-hospital mortality among patients admitted to the critical care units (CCUs) in Vietnam.

**Design** Retrospective study.

**Setting** The whole 1-year data of admissions to CCUs were collected from 34 hospitals from January to December 2018.

**Participants** A total of 44 013 episodes of admission to CCUs were analysed.

**Primary outcome** We used International Classification of Diseases-11 codes to assess the primary diagnosis associated with admissions and in-hospitals mortality. Years of life lost (YLL) measure was further used to estimate the burden of disease.

**Results** The 0–5 years and ≥70 years age groups accounted for 14.8% (6508/44 013) and 26.1% (11 480/44 013) of all admissions, respectively. The most common diagnoses were diseases of the respiratory system (27.8% or 12 255/44 013), followed by unclassified symptoms, signs or clinical findings (13% or 5712/44 013), and diseases of the circulatory system (12.2% or 5380/44 013). Among 28 311 patients with available outcome data, 1681 individuals (5.9%) died during the hospitalisation. The in-hospital mortality rate increased with age, from 2.8% (86/3105) in under 5 years old age group to 23.1% (297/1288) in over 90-year age group. Diseases of the respiratory system was the leading causes of death in term of number of deaths (21.8% or 367/1681 of all deaths). Diagnosis of sepsis was associated with the highest in-hospital mortality (36.8%). The overall YLL under the age of 75 were 1287 per 1000 patients.

**Conclusions** CCUs in Vietnam faced wide differences in the burden of diseases. Sufficient infrastructure and adequate multidisciplinary training are essential to ensure the appropriate response to the current needs of population.

## INTRODUCTION

Emergency care is an essential service of the hospitals which serves as a first point of contact to provide immediate treatment for individual with acute illness and life-threatening needs. Over 50% of mortality and over 40%

### STRENGTHS AND LIMITATIONS OF THIS STUDY

⇒ We conducted a large multicentre study on the pattern of admission and frequency of diagnosis in the critical care units in Vietnam, a lower-middle-income country.

⇒ We present mortality, cause of admission and the years of life lost due to premature death in critical care units in provincial and district hospitals in Vietnam.

⇒ The major limitation relates to the accuracy of the diagnosis and diagnostic coding in the study hospitals.

of all morbidities worldwide can be contributed by emergency medical conditions.[1] In May 2019, the 63rd World Health Assembly recognised that existing data do not provide adequate support for effective planning and resource allocation for emergency care and urged member states to identify the local acute illness burden for improving quality of emergency care.[2]

The International Statistical Classification of Diseases and Related Health Problems (often referred to as the International Classification of Diseases, ICD) has been used in more than 120 countries to monitor the incidence and prevalence of diseases.[3] Vietnam has implemented the ICD since 1998 as a tool for reporting morbidity, cause of death and billing for health insurance in hospitals. The burden of disease in Vietnam has changed rapidly over the past two decades with a marked shift towards the predominance of non-communicable diseases. Between 2000 and 2017, the share of communicable disease has reduced a half (24%–14%), the share of non-communicable disease has increased almost 10% (63%–74%) while the share of injuries still remained stable at around 12%–13% (all measured in the disability-adjusted life-year

(DALY)).[4] However, the research specifically measuring the burden of diseases in emergency departments (EDs) or intensive care capable EDs (referred to as critical care units or CCUs) in Vietnam has been limited. Recently, some studies on the pattern of hospitalisation, the causes of death or the hospital death certification have been performed,[5–7] but none of these studies has provided a comprehensive description of the patterns of diagnosis or measured the premature death by the years of life lost (YLL) metric in patients with admission to CCUs in the country.

The lack of infrastructure and human resource brings significant challenges to the delivery of critical care in low-income and middle-income countries (LMICs).[8] The emergence of COVID-19 has put further pressure on the critical care system in these countries that already struggling with the high burden of admission to the CCUs in the pre-COVID-19 era. An understanding of the conditions that involve an admission to CCU will allows policymakers and planners to implement interventions that are effective in the national contexts to meet the care quality needs of the rapidly expanding ageing populations and to prevent the disruption of the critical care services in the pandemic. Our aim was to describe the frequency of diagnostic categories, in-hospital mortality, cause of death and the burden of premature mortality measuring by YLL in CCUs in Vietnam in 2018.

## METHODS
### Study design and participants
This was a retrospective study of all episodes of CCU admissions to selected provincial and district hospitals in Vietnam from 1 January 2018 to 31 December 2018. We purposely selected 5 out of 63 provinces to represent 5 geographical areas in Vietnam. In each province, we invited all CCUs in hospitals in the catchment area.

Data were collected from the electronic hospital management system, including patient's demographic information, diagnosis, dates of admission, length of stay and outcome at discharge. Diagnosis of individual patients in each episode of admission were originally recorded in the ICD, 10th revision (ICD-10) codes. Because the ICD, 11th revision (ICD-11) came into effect in January 2022, for the monitoring of any change in the pattern of diseases responsible for admission in the future, all ICD-10 codes were translated to the ICD-11 codes (version 05/2021) using WHO mapping tables with the correspondence between ICD-10 and ICD-11 codes and link the ICD-11 codes with their corresponding associated descriptive labels. Two independent investigators (VQD and BTKL) manually reviewed any non-unique code or discrepancies in mapping between ICD-10 and ICD-11 which were eventually resolved by the consensus of two reviewers with the reference to the majority of code assignments by treating doctors.

The outcomes from the hospitalisation were defined as death in patients who died in the hospital or was going

home to die imminently after hospital discharge (palliative discharge). Palliative discharge is a common practice in Vietnam when a patient is in a moribund state and expresses a wish to die at home. Deterioration was defined as any worsening of clinical conditions by the treating doctor's judgement at the time of discharge. The data on the outcomes at CCU discharge and interventions during CCU stay were not available for analysis.

### Statistical analysis
For the analysis of pattern of admission, we included all episodes of admission. Consequently, one patient might have multiple episodes of admission to CCUs during a year. For each episode, the primary diagnosis which was defined as the main reason for CCU admission was counted. The primary diagnosis for that episode may be the same or different from the previous episode of admission. Individual patients were identified within the datasets by matching on full name, gender, date of birth and insurance numbers when they were available). The in-hospital mortality was calculated by dividing the number of deaths by the total number of individual patients.

YLL is a measure of premature mortality which is calculated from the number of deaths at each age multiplied by a standard life expectancy for the age at which death occurs.[9] Vietnam's standard life expectancy at birth in 2019 for both sexes were 73.7 years old[10] and to compare with other studies, we considered premature death were deaths under 75 years old. We also calculated the average YLL by dividing total of YLL by the number of premature death and the crude YLL rate by dividing total of YLL by the total number of patients aged under 75 years (expressed per 1000 persons).

We presented categorical variables as frequencies and percentages, while continuous variables as medians and IQR. We used the $\chi^2$ test or Fisher's exact test to compare two binomial proportions. Statistical analyses were performed using SPSS V.27 (IBM). Significant differences were considered if two-sided p values were <0.05.

### Patient and public involvement
Patients and the public were not involved in the design, recruitment or conduct of the study.

## RESULTS
Between 1 January 2018 and 31 December 2018, there were 44 013 admission episodes to CCUs from 34 hospitals in 5 provinces (namely Hanoi, Thai Nguyen, Ha Nam, Kon Tum and Can Tho), corresponding to 38 534 unique patients. There were 3647 (9.5%) patients with two or more admission in the study year (figure 1). The median age was 53 years (IQR) 25–70) and included 53.4% male. The majority of CCU admissions in this study (31 569, 71.7%) were at the district hospitals. The median length of stay was 5 (IQR 2–8) days. A total of 1764/36 518 admissions (4.8%) had a length of stay of more than 14 days. The hospitalisation outcome was available for 28

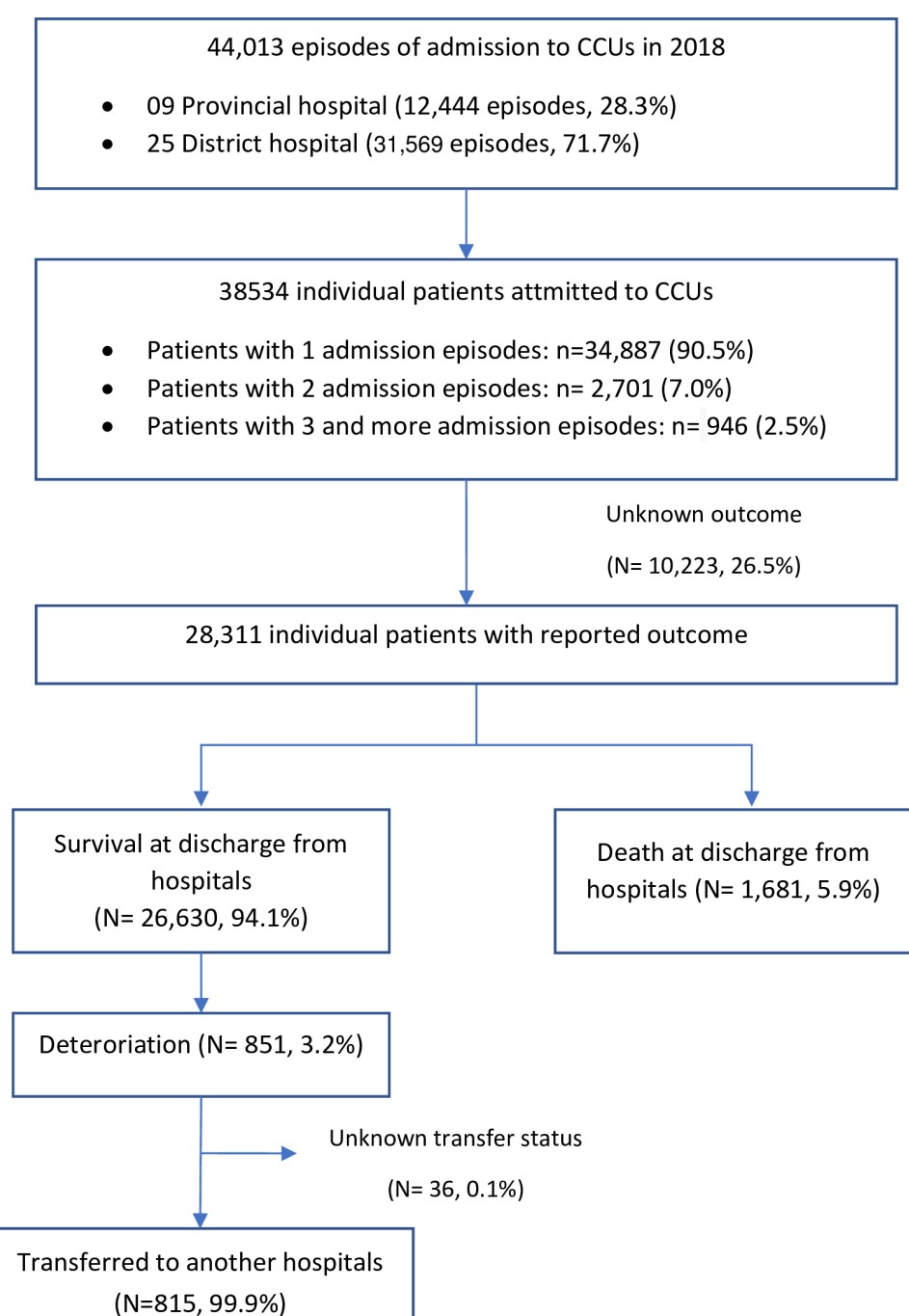

**Figure 1** Flow chart of CCU admission and outcome. CCU, critical care unit.

311/38 534 (73.5%) patients. Overall survival to hospital discharge was 94.1% (26 630/28 311).

### Admission diagnosis in CCUs

Of 44013 episodes of admission to CCUs, the most common diagnosis were respiratory diseases (12 255 episodes, 27.8%) in which, 38.7% was certain lower respiratory tract diseases (blockL1-CA2) and 34.4% was lung infections (blockL1-CA4), followed by unclassified symptoms, signs or clinical findings (5712 episodes or 13.0%) (51.8% was of digestive system or abdomen, blockL1-MD8), circulatory system diseases (5380 episodes,

12.2%) (51,2% was hypertensive diseases, blockL1-BA0), diseases of the nervous system (5019 episodes, 11.4%) (85.4% was cerebrovascular diseases, blockL1-8B0) and injury, poisoning or consequences of external causes (4186 episodes or 9.5%) (39.7% was harmful effect off substances, blockL1-NE6). The percentage of diagnosis in episodes of CCU admission varied by age groups (figure 2). The number of admissions peaked in children from 5 years old and younger and was lowest in the 5–10 years age group (figure 3). The respiratory disease was the leading cause of admission in the under 10 year age

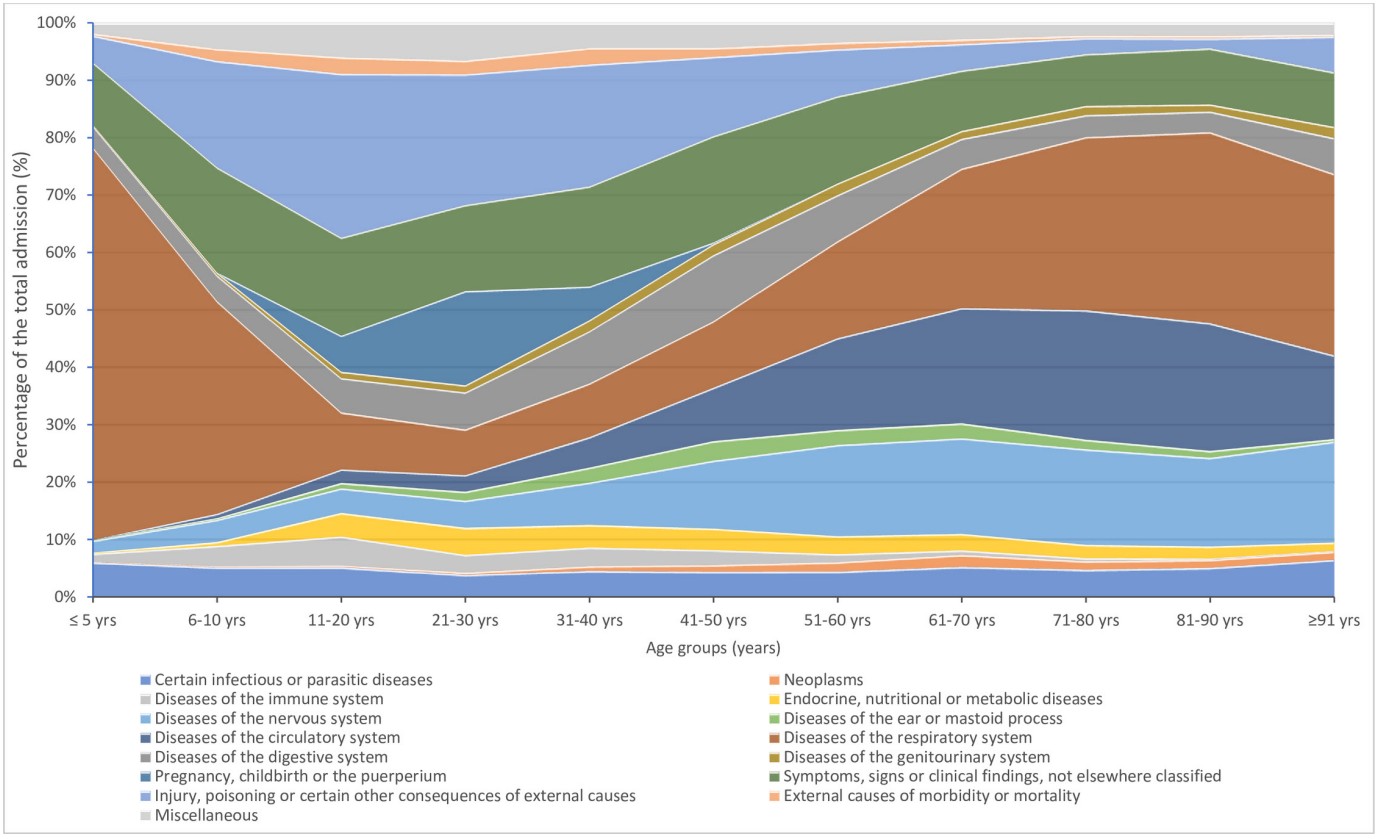

**Figure 2** Diagnosis on admission to CCU by age groups. CCU, critical care unit.

group (64% of admissions) and age groups of 61 years and older (29% of admissions) (figure 2).

The number of admissions in all study hospitals fluctuated across months but mostly driven by the admission by respiratory diseases (online supplemental figure 1). The proportions of admission to CCUs were significantly higher in provincial hospitals than in district hospital in regard to diseases of the nervous system (18.7% vs 8.5%, p<0.001), certain infectious or parasitic diseases (8.9% vs 3.4%, p<0.001), injury/poisoning (10.3% vs 9.2%, p=0.00), endocrine, nutritional or metabolic diseases (3.6% vs 2.2%, p<0.001) and neoplasms (2% vs 0.7%, p<0.001) (figure 4).

### Outcome of hospitalisation

The median length of stay for all admissions was 5 days (IQR 2–8). Among 28 311 patients with available outcome data and, 1681/28 311 (5.9%) died. The median age of death was 68 years (IQR 47–86) and 63.3% of death were in men. Men had significantly higher in-hospital mortality rate than women (3.8% vs 2.6%, p<0.001). The median duration of hospital stay for the survivors and death cases was 4 (IQR:2–8) and 3 (IQR: 2–7), respectively. More than three quarters of deaths (1290/1681 or 76.7%) occurred in provincial hospitals. The all-cause in-hospital mortality rates by age groups and mortality rate for selected groups of diagnosis are presented in figure 3 and online supplemental table 1. The most common causes of death by

ICD-11 block codes and chapter codes are showed in table 1 and online supplemental table 2, respectively.

The in-hospital mortality in patients over 75 years was 662/6,140 (10.8%). In total, 28 135 000 years were lost among patient admitted to the study CCUs. This yielded an overall YLL under the age of 75 years of 1287 per 1000 people. Table 1 presents the top 10 causes of death by ICD-11 block codes. Cerebrovascular diseases rank the first in the number of deaths (303/1681 or 18%) and the 11 when measuring in in-hospital mortality (8.7%) while sepsis ranked as the first specified cause of death in terms of in-hospital mortality (36.8%) and ranked fourth in term of death counts.

### DISCUSSION

This study presents for the first time, data to quantify the burden of diseases in CCUs, for the most common reasons of admission and the in-hospital mortality of patients admitted to CCUs in Vietnam. Few studies on admission to the CCUs in LMICs have been reported earlier.

Burden of diseases in Vietnam has changed significantly in the past 20 years.[4] The communicable diseases burden (measured in DALYs) decreased rapidly from 37% in 1990 to 14% in 2017. However, in the critical care settings, our study shows that the infectious diseases remain a major emergency medical condition and are the prime cause of

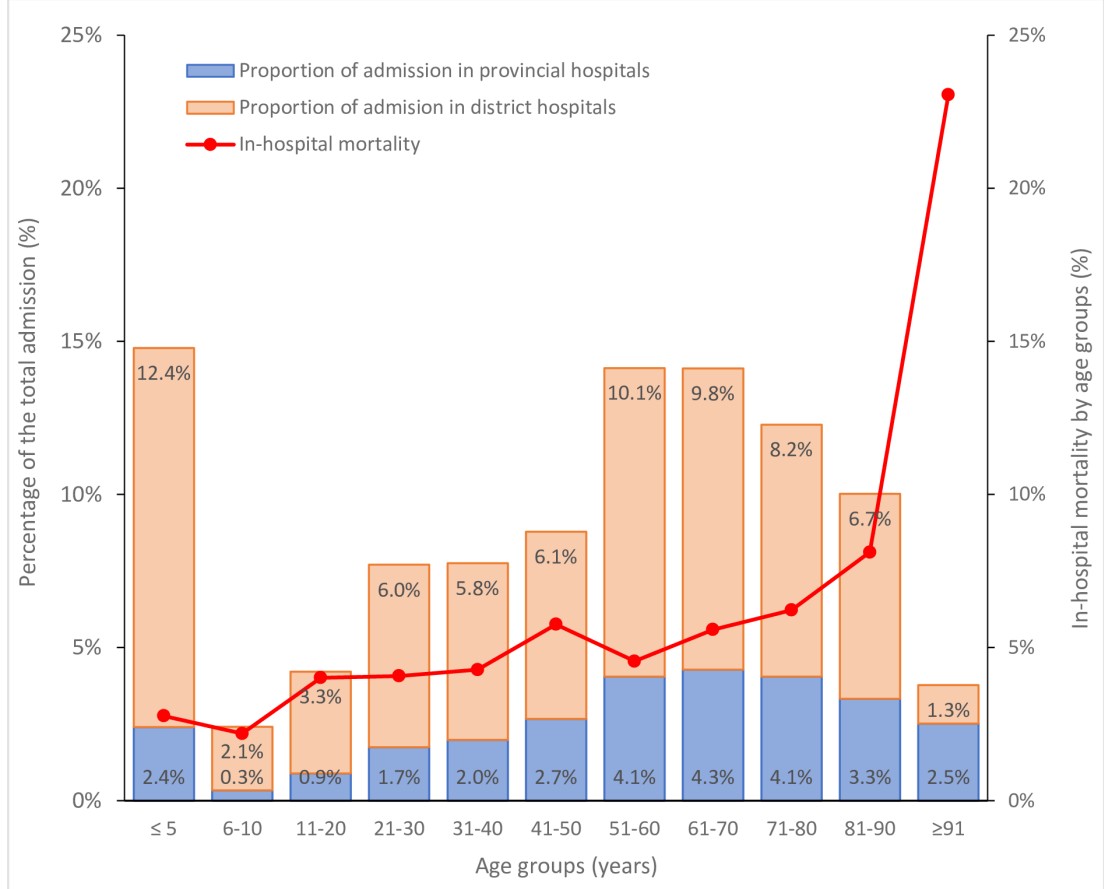

**Figure 3** Number of admission and In-hospital mortality by age groups.

morbidity and mortality in terms of number of admission and in terms of YLL.

Our findings on the difference distribution of diseases in CCUs in provincial and district hospitals reflected the variation of healthcare services and capacity of clinical management. In 2019, we conducted a survey on the capacity and use of diagnostics and treatment for patients

with severe acute respiratory infections in 48 hospitals in Vietnam (including 32 hospitals in this study) at the same time as this study. Significant differences were observed in the district hospitals when compared with provincial hospitals in regard to availability of the microbiological diagnostics (9.4% vs 81.2%, p<0.001), inflammatory markers of C reactive protein (31.2% vs 54.2%,

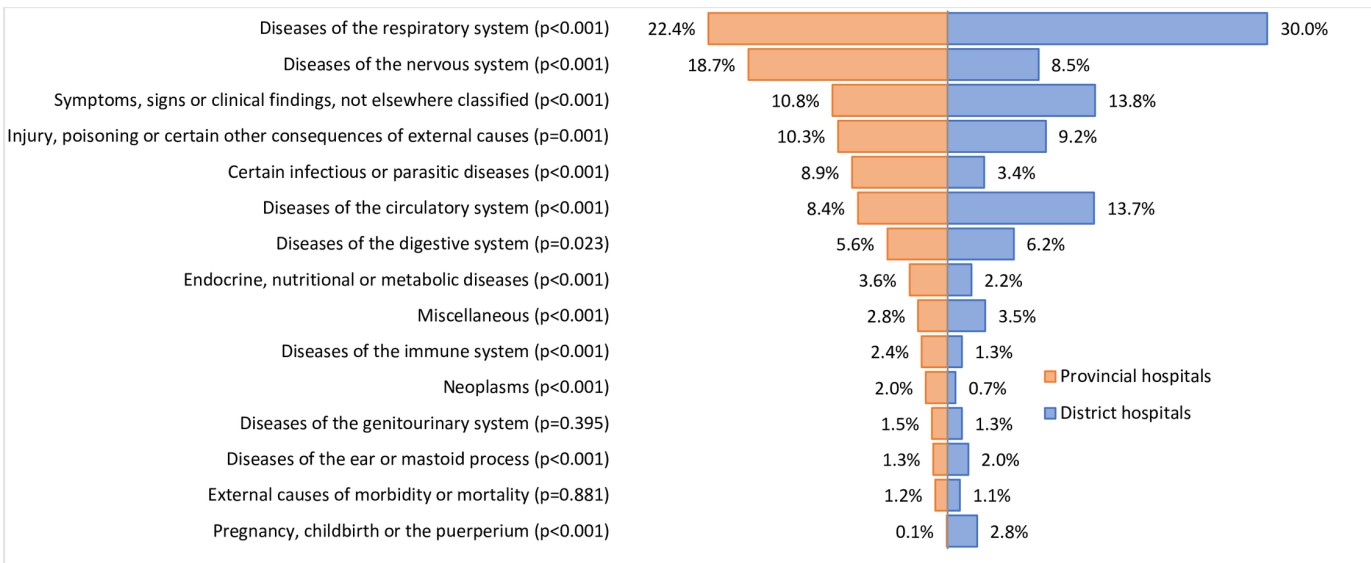

**Figure 4** Cause of admission to CCUs by level of care. CCU, critical care unit.

Table 1  The top 20 causes of death in terms of years of life lost (YLL) and numbers of deaths by ICD-11 block level one codes

| Diagnosis | No and proportion of death cases | In-hospital mortality (n,%) | Total YLL75 | Proportion of YLLs75 | YLL75 per 1000 patients | Length of stay to death (median, IQR) |
|---|---|---|---|---|---|---|
| Cerebrovascular diseases | 303 (18.0%) | 8.7% | 3 211 000 | 12.3% | 1380 | 3 (2–6) |
| Miscellaneous | 259 (15.4%) | 2.3% | 5 686 000 | 38.2% | 614 | 2 (1–6) |
| Lung infections | 201 (12.0%) | 7.2% | 2 875 000 | 9.5% | 1440 | 7 (3–15) |
| Sepsis | 149 (8.9%) | 36.8% | 1 991 000 | 2.6% | 8127 | 4 (2–11) |
| Harmful effects of substances | 87 (5.2%) | 6.1% | 3 534 000 | 5.0% | 2544 | 1 (1–2) |
| Respiratory failure | 79 (4.7%) | 18.1% | 365 000 | 1.6% | 1714 | 9 (5–15) |
| General symptoms, signs or clinical findings | 67 (4.0%) | 13.5% | 1 507 000 | 2.2% | 3712 | 2 (1–4.5) |
| Injuries to the head | 66 (3.9%) | 15.6% | 1 140 000 | 1.6% | 3167 | 2 (2–4) |
| Certain lower respiratory tract diseases | 63 (3.7%) | 2.1% | 443 000 | 10.2% | 217 | 4 (2–10) |
| Symptoms, signs or clinical findings of the digestive system or abdomen | 61 (3.6%) | 5.5% | 994 000 | 4.0% | 1062 | 3 (2–6) |
| Malignant neoplasms, except primary neoplasms of lymphoid, haematopoietic, central nervous system or related | 50 (3.0%) | 18.0% | 587 000 | 1.0% | 3073 | 6 (3–9) |
| Symptoms, signs or clinical findings of the circulatory system | 50 (3.0%) | 31.8% | 1 060 000 | 0.7% | 8217 | 2 (1–6) |
| Heart failure | 39 (2.3%) | 8.4% | 280 000 | 1.6% | 1466 | 3 (1–5) |
| Ischaemic heart diseases | 37 (2.2%) | 5.2% | 78 000 | 2.5% | 158 | 3 (1–5) |
| Diseases of liver | 36 (2.1%) | 11.2% | 549 000 | 1.1% | 1861 | 2.5 (1.5–7.5) |
| Injuries involving multiple body regions | 29 (1.7%) | 28.2% | 439 000 | 0.4% | 6362 | 2 (1–6) |
| Symptoms, signs or clinical findings of the respiratory system | 28 (1.7%) | 8.4% | 706 000 | 1.1% | 3017 | 2 (1–4) |
| Mental or behavioural symptoms, signs or clinical findings | 22 (1.3%) | 13.3% | 363 000 | 0.8% | 2771 | 2 (1–2) |
| Allergic or hypersensitivity conditions | 20 (1.2%) | 3.3% | 890 000 | 2.2% | 1537 | 2 (1–2) |
| Infection, unspecified | 19 (1.1%) | 31.7% | 741 000 | 0.2% | 14 529 | 2 (2–4) |
| Epilepsy or seizures | 16 (1.0%) | 5.0% | 696 000 | 1.1% | 2275 | 1.5 (1–2) |
| Total | 1681 (100.0%) | 5.9% | 28 135 000 | 100.0% | 1287 | 3 (2–7) |

ICD, International Classification of Disease.

p<0.001) and procalcitonin (3.1% vs 25%, p<0.001) and mechanical ventilation (40.6% vs 60.4%, p<0.001).[11] Provincial hospitals in Vietnam are the secondary hospitals which do not only treat referred patients from lower level of care (district hospitals or commune centres) but also admit patients from the community. Additionally, patient can directly self-refer to higher level of hospitals which resulting in under-utilisation of district hospitals and overcrowding in provincial hospitals, especially for given specialities which requires higher levels of skills and capacity.

Our findings are partly aligned with previous reports in all-cause mortality in both sexes and all ages showing that cardiovascular (ischaemic heart disease, stroke), respiratory diseases and injuries were the significant challenge. WHO's report in 2019 showed that ischaemic heart disease, stroke, neonatal conditions, chronic obstructive pulmonary disease (COPD) and lower respiratory infections were the top five causes of death in term of death counts in LMICs.[12] At the global level in 2019, the top three diseases with highest burden (measuring in DALY) were neonatal disorder (32.9% of global DALYs), lower respiratory infections (11% of global DALYs) and diarrhoeal diseases (9.3% of global DALYs) in children younger than 10 years and were ischaemic heart diseases (16.2% of global DALY), stroke (13% of global DALY) and COPD (8.5% of global DALY) in patients from 75 years and older.[13] In Vietnam, stroke (164.8 deaths per 100 000), ischaemic heart disease (95.3 deaths per 100 000), COPD (37.2 deaths per 100 000) and diabetes mellitus (35.1 deaths per 100 000) and road injuries (30.5 deaths per 100 000) were in the top-five rankings for total

number of death in 2019.[14] Sepsis and harmful effects of substances were not listed in the top five causes of death in the general population, but we found that these were important conditions in emergency care settings.

The overall mortality in our study was higher than previous report. In a systematic review of data from 65 EDs in 59 LMICs countries between 2013 and 2014, the median mortality was 1.8% (IQR 0.5%–5.1%), highest in sub-Saharan Africa (median 3.4%, IQR: 0.5%–6.3%) and lowest in South Asia, East Asia and Pacific (0.3, IQR 0.2%–0.8%).[15]

In our study, patients aged 75 years or older presented 18.9% of all admission at CCUs despite the fact that they comprised 3.2% of general population.[16] The percentage of population older than 75 years in Vietnam is expected to increase to 4.2% by 2030 and 6.6% by 2040.[16] This will increase the demand for critical care resources and requires further evaluation to guide healthcare decision-making and changes in organisational models in CCU and delivery of critical care services during a pandemic.

The design of multicentre study, large population and data collection for the whole year of 2018 are the strengths of our study. The major limitations related to the diagnosis description and diagnostic coding by doctors in study hospitals. Data quality of diagnosis may vary between hospitals due to the accurate clinical diagnosis of disease depends on doctor's knowledge and skills, available laboratory investigations and diagnostic imaging procedures. However, there are differences in the capacity of clinical management and use of diagnostics between provincial and district hospitals in the country.[11] In Vietnam, treating doctors are responsible for assigning the ICD diagnostic code to each patient discharged from the hospital. The complex nature of ICD coding, patient's condition in critical care settings and the lack of coding experience pose further challenges for the accuracy and reliability of the diagnosis given across hospitals. Another limitation was the missing information on outcome for 26.5% (10 223/38 543) of patients because these were not actually entered into the electronic hospital management system. These missing data can cause bias in the estimation of in-hospital mortality.

Currently, there is limited cost-effectiveness analysis performed for critical care services in Vietnam. The results of this study in regard to the variety of diagnoses and outcomes of admission will direct the future studies on the establishment of CCU registry, the effectiveness of CCU treatments and estimates of the total cost to save a life-year. This information will be helpful to develop policies and strategies to make better-informed decisions in resource-constrained settings in the face of an ageing population in Vietnam.

## Conclusion

Our study indicates a variation of needs of emergency care by age groups and levels of care among patients admitted to CCUs in provincial and district hospitals in Vietnam. These findings support the need to provide multidisciplinary team training for clinicians in CCUs and the future studies should further assess the emergency and critical care capacity needs, ability of the healthcare setting to provide the emergency services and the quality of care (including minimal set of indicators) and gaps to mobilise the resources required for live saving during a pandemic.

**Acknowledgements** We are very grateful to the patients and staff at the participating hospitals in Ha Noi, Thai Nguyen, Ha Nam, Kon Tum and Can Tho. We would like to thank Dr Satoko Otsu and Vu Quang Hieu (WHO, Vietnam Country Office) for their tireless support and contribution to strengthening the capacity for clinicians in Vietnam and thank Dr Nguyen The Hung (National Hospital for Tropical Diseases) for his initial support in the implementation of this study.

**Contributors** VQD conceived and designed the study. VQD analysed the data and wrote the first draft of the report. VQD, BTKL and GBK involved to the acquisition and interpretation of data. All authors contributed to and approved the final report. VQD is guarantor.

**Funding** This work was supported by Viet Nam-WHO Country Office.

**Disclaimer** The funder of the study had no role in study design, data collection, data analysis, data interpretation, or writing of the report.

**Competing interests** None declared.

**Patient and public involvement** Patients and/or the public were not involved in the design, or conduct, or reporting, or dissemination plans of this research.

**Patient consent for publication** Not applicable.

**Ethics approval** This study was approved by the institutional review board (IRB) in the Hanoi Medical University (59/GCN-DDNCYSH-DHYHN).

**Provenance and peer review** Not commissioned; externally peer reviewed.

**Data availability statement** Data are available on reasonable request.

**ORCID iDs**
Vu Quoc Dat http://orcid.org/0000-0002-5904-5970
Bui Thi Khanh Linh http://orcid.org/0000-0002-7337-5925

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
