## [Reviewer comments · BMJ Open]

ARTICLE DETAILS

TITLE (PROVISIONAL)	Causes of admissions and in-hospital mortality among patients admitted to critical care units in primary and secondary hospitals in Vietnam in 2018: a multicentre retrospective study
AUTHORS	Dat, Vu Quoc; Linh, Bui Thi Khanh; Kim, Giang

VERSION 1 – REVIEW

REVIEWER	Watters, David Deakin University, Surgery
REVIEW RETURNED	14-Mar-2022

GENERAL COMMENTS	1 Ethics: There is no statement about ethics that I could find in the methods. I recognise that you have interrogated administrative data collected for the purpose of this study. It may not be necessary to have ethics approval for what is essentially an audit, but if not required in Vietnam please say so. 2 The paper is nicely contextualised within the global burden of disease study and the use of YLL is a valuable addition to the literature on burden of disease in Vietnam. 3 Because of its reliance on coding the paper is a little light on clinical conditions, particularly postoperative critical care, which is quite influential on the outcomes of patients treated in CCU's. This information would be helpful. Given the authors have the codes, I wonder whether a table of actual conditions (eg I can see head injuries but I can't see acute abdominal pathology/post emergency laparotomy, or specific fractures within the injury list). Could some of the system diagnoses have supplementary tables that cover actual conditions as opposed to broad groups? 4 Figure 1: looking at the figure and reading its legend it is not clear whether the survival or death at discharge is survival or death at discharge from ICU or hospital. It would be good to present both if available, otherwise death at discharge from hospital is the most valuable which is what I think Fig 1 refers to given your statement on this in the methods section. 5 Similar to my comment on emergency laparotomy, it is not clear to me from the paper what treatments were available. One assumes a CCU has ventilation, inotropes, haemofiltration available, but perhaps this is not true in all district hospitals. 6 If you are going to limit this paper to the burden of disease your discussion needs to discuss treatment facilities, and also progress to treatment cost-effectiveness. A follow up paper to recommend would be the cost effectiveness of CCU in Vietnam - how much did it cost to save a life year or 10 life years? Are there recommendations to be made around improving facilities?
---

REVIEWER	Kim, Won Young
-----------------	----------------

	Asan medical center, emergency medicine
REVIEW RETURNED	13-Apr-2022

GENERAL COMMENTS	Authors were to describe the frequency of diagnostic categories, in-hospital mortality, cause of death and the burden of premature mortality in CCUs of Vietnam in 2018. In general, the context and motivation for this research is relatively weak, and the problem to be solved by the research is relatively vague. These are just descriptive statistics. Although the information appears to be rich, the accuracy and reliability of diagnosis description or diagnostic coding is limitation. In the conclusions section, representation should be linked to the research findings that served.  1. This study mainly focuses on the diagnosis, mortality, and causes of death in patients admitted to CCU. Were the patients included regardless of the route of hospitalization? (e.g., admission from outpatient department, or emergency department, or general ward) If they were, it may be appropriate to revise the wording of emergency department and emergency care in the introduction and conclusion section. Current tone may confuse readers that this study focuses on health care in the emergency department. 2. What is the definition of “going home to die imminently after hospital discharge”? Please describe the definition in the methods section. 3. Please describe the definitions of provincial and district hospital in the methods section. And It may be more appropriate to describe functional differences between the hospitals in the methods section, rather than in the discussion section. 4. Approximately 40% of patients were missing from the outcome analysis. Considering the purpose of this research, it is important to investigate the outcomes of the included patients. A large number of missing values could compromise the integrity of conclusions. Please explain the main reasons of missing data in the methods section. If the mortality rate has increased compared to previous studies due to the missing data, please describe it as a study’s limitation also. 5. In table 1, which ICD codes were included in “miscellaneous” category? It should be described or discussed in detail because it is the second largest proportion of death cases. 6. The authors presented data as background information that the burden of non-communicable disease has increased since 1990. The reduction of communicable disease may depend on changes in the medical/social environment such as available resource, health care policy, etc. However, the authors concluded that the preparedness for infectious diseases is important as they are increasing. Please describe why the authors mentioned the increase in non-communicable diseases at the outset. 7. The authors concluded that respiratory and infectious diseases became a high burden of illness in emergency care in Vietnam. However, this interpretation may vary depending on the data presented. In terms of block level 1 code, cerebrovascular diseases and sepsis show the highest number of death and mortality, respectively (table 1). Otherwise, in terms of disease
--

	classification, respiratory diseases and neoplasms show the most (supp table 1). I think that the authors should revise the conclusion section reflecting those differences. 8. There are no specific proposals or suggestions derived from the results of current study. Please add a paragraph in the discussion section about the application of the research finding in clinical practice and future research to compliment the limitations of current study. 9. In the conclusion section, the authors mentioned that the medical needs varied by age group, but a detailed interpretation of the conclusion was not included in the discussion section. Fig 2 and 3 look fancy with color graphs, however, they only show the obvious and expected results of this study. Please add more detailed interpretations of the distribution of diagnoses and mortality by age groups in the discussion section. 10. What is the meaning of supplementary Figure 1 (Pattern of admissions over months). It is not easy to follow. Please add the labels of X-axis and Y-axis and interpret this figure in discuss section
--	---

VERSION 1 – AUTHOR RESPONSE

Reviewer: 1

Dr. David Watters, Deakin University, Barwon Health

Comments to the Author:

1 Ethics: There is no statement about ethics that I could find in the methods. I recognise that you have interrogated administrative data collected for the purpose of this study. It may not be necessary to have ethics approval for what is essentially an audit, but if not required in Vietnam please say so.

Our response: We recognized the importance of the ethics aspect and stated that “This study was approved by the institutional review board (IRB) in the Hanoi Medical University (59/GCN-DDNCYSH-DHYHN)”. Please see the “Ethics approval” section.

2 The paper is nicely contextualised within the global burden of disease study and the use of YLL is a valuable addition to the literature on burden of disease in Vietnam.

Our response: Thank you.

3 Because of its reliance on coding the paper is a little light on clinical conditions, particularly postoperative critical care, which is quite influential on the outcomes of patients treated in CCU's. This information would be helpful. Given the authors have the codes, I wonder whether a table of actual conditions (eg I can see head injuries but I can't see acute abdominal pathology/post emergency laparotomy, or specific fractures within the injury list).

Could some of the system diagnoses have supplementary tables that cover actual conditions as opposed to broad groups?

Our response: Thank you, we have provided the details of the actual conditions using ICD-11 4 character codes in death cases in the Supplementary table 2.

4 Figure 1: looking at the figure and reading its legend it is not clear whether the survival or death at discharge is survival or death at discharge from ICU or hospital. It would be good to

present both if available, otherwise death at discharge from hospital is the most valuable which is what I think Fig 1 refers to given your statement on this in the methods section.

Our response: Thank you. We provided the data on the survival and death at discharge from hospital. We have revised the text in the figure 1 for the clarity.

5 Similar to my comment on emergency laparotomy, it is not clear to me from the paper what treatments were available. One assumes a CCU has ventilation, inotropes, haemofiltration available, but perhaps this is not true in all district hospitals.

Our response: Thank you for pointing that out. We agree that it is important to understand the availability of diagnostics and treatment. At the time collecting the data, we conducted a concurrent survey of capacity and use of diagnostics and treatment for patients with severe acute respiratory infections in Viet Nam. There were significant differences between district and provincial hospitals in the availability of microbial culture, rapid influenza diagnostic tests, inflammatory markers and mechanical ventilation. We published the survey results at <https://www.ncbi.nlm.nih.gov/pmc/articles/PMC8873919/>. As your advised, we have added the below texts about these differences and cited the above-mentioned article in the discussion.

“In 2019, we conducted a survey on the capacity and use of diagnostics and treatment for patients with severe acute respiratory infections in 48 hospitals in Vietnam (including 32 hospitals in this study) at the same time as this study. Significant differences were observed in the district hospitals when compared to provincial hospitals in regards to availability of the microbiological diagnostics (9.4% vs 81.2%, $p < 0.001$), inflammatory markers of C-reactive protein (31.2% vs 54.2%, $p < 0.001$) and procalcitonine (3.1% vs 25%, $p < 0.001$) and mechanical ventilation (40.6% vs 60.4%, $p < 0.001$)”

6 If you are going to limit this paper to the burden of disease your discussion needs to discuss treatment facilities, and also progress to treatment cost-effectiveness. A follow up paper to recommend would be the cost effectiveness of CCU in Vietnam - how much did it cost to save a life year or 10 life years? Are there recommendations to be made around improving facilities?

Our response: Thank you, this comments is important and helpful to the study team for future study. We would like to incorporate your comment in our manuscript as below:

“Currently, there is limited cost-effectiveness analysis performed for critical care services in Vietnam. The results of this study in regards to the a variety of diagnoses and outcomes of admission will direct the future studies on the establishment of CCU registry, the effectiveness of CCU treatments and estimates of the total cost to save a life-year. This information will be helpful to develop policies and strategies to make better-informed decisions in resource-constrained settings in the face of an aging population in Vietnam.”

Reviewer: 2

Dr. won young kim, Asan medical center

Comments to the Author:

Authors were to describe the frequency of diagnostic categories, in-hospital mortality, cause of death and the burden of premature mortality in CCUs of Vietnam in 2018. In general, the context and motivation for this research is relatively weak, and the problem to be solved by the research is relatively vague. These are just descriptive statistics. Although the information appears to be rich, the accuracy and reliability of diagnosis description or diagnostic coding is limitation. In the conclusions section, representation should be linked to the research findings that served.

1. This study mainly focuses on the diagnosis, mortality, and causes of death in patients admitted to CCU. Were the patients included regardless of the route of hospitalization? (e.g.,

admission from outpatient department, or emergency department, or general ward) If they were, it may be appropriate to revise the wording of emergency department and emergency care in the introduction and conclusion section. Current tone may confuse readers that this study focuses on health care in the emergency department.

Our response: Thank you, in our study, we included all patients regardless of the route of hospitalization. According to Vietnam government's regulations, an emergency department (ED) is a medical treatment facility which admits and treats all patients with emergency conditions referred to the hospital. Its function is to assess, triage and provide appropriate management by priority level of emergency until the patient is no longer in a serious condition and then within 48 h must transfer the patient to an Intensive care unit (ICU) or an appropriate medical ward when patient's status allow. An ICU is a clinical department responsible for providing continuous critical care for patients who are transferred from an ED or clinical wards of the hospital. Because many provincial and district hospitals cannot establish separated ED and ICU, we used the terms critical care unit (CCU) is referred to emergency departments (EDs) or intensive care capable EDs as we briefly described in the introduction.

2. What is the definition of “going home to die imminently after hospital discharge”? Please describe the definition in the methods section.

Our response: Thank you, we have added the definition for the clarity. Palliative discharge is a common practice in Vietnam, patients prefer to die at home. Thus when they are in a moribund state they often request discharge. As this is a retrospective study of routine care we are not able to comment on whether such patients did indeed pass away, however from experience it is unlikely that these patients remained alive for any length of time. Palliative discharges have been considered as inpatient deaths in a number of previous studies from this setting by our group and others (Dat et al. BMC Infect Dis doi: 10.1186/s12879-017-2582-7; Miles et al. PLoSOne 2017 doi.org/10.1371/journal.pone.0173407). We have added the below text “Palliative discharge is a common practice in Vietnam when a patient is in a moribund state and expresses a wish to die at home”.

3. Please describe the definitions of provincial and district hospital in the methods section. And It may be more appropriate to describe functional differences between the hospitals in the methods section, rather than in the discussion section.

Our response: Thank you, we agree that it is important to describe the functions of difference level of care. However, we would like to describe these differences next to the discussion of variation of disease by levels of hospital to help readers having some ideas about the reason why there were difference in causes of admission in district and provincial hospitals.

4. Approximately 40% of patients were missing from the outcome analysis. Considering the purpose of this research, it is important to investigate the outcomes of the included patients. A large number of missing values could compromise the integrity of conclusions. Please explain the main reasons of missing data in the methods section. If the mortality rate has increased compared to previous studies due to the missing data, please describe it as a study's limitation also.

Our response: we would like to confirm that only 26.5% (10,223/38543) of patients were missing from the outcome analysis (figure 1). We recognized that the *missing data* can produce *biased*. We have revised the discussion of limitation as below:

“Another limitation was the missing information on outcome for 26.5% (10,223/38543) of patients because these were not actually entered into the electronic hospital management system. This missing data can cause bias in the estimation of in-hospital mortality”

5. In table 1, which ICD codes were included in “miscellaneous” category? It should be described or discussed in detail because it is the second largest proportion of death cases.

Our response: Thank you, it is also a comment from another reviewer. We have added the appendix to provide the details of miscellaneous causes (the supplementary table 2).

6. The authors presented data as background information that the burden of non-communicable disease has increased since 1990. The reduction of communicable disease may depend on changes in the medical/social environment such as available resource, health care policy, etc. However, the authors concluded that the preparedness for infectious diseases is important as they are increasing. Please describe why the authors mentioned the increase in non-communicable diseases at the outset.

Our response: Thank you for this comment. In general, the burden of non-communicable decreased as measured in the DALY between 2000 and 2017. However, our analysis showed that the burden of infectious diseases was significant in CCUs in term of number of admission (27.8% of all admission episodes) and this measure was not well reported before. It reflected the fact that a single measure is not enough. We described the admission in Vietnam in 2018 and in the context of the emergence of COVID-19, we believed that the preparedness for the threats of infectious diseases in the future is important.

7. The authors concluded that respiratory and infectious diseases became a high burden of illness in emergency care in Vietnam. However, this interpretation may vary depending on the data presented. In terms of block level 1 code, cerebrovascular diseases and sepsis show the highest number of death and mortality, respectively (table 1). Otherwise, in terms of disease classification, respiratory diseases and neoplasms show the most (supp table 1). I think that the authors should revise the conclusion section reflecting those differences.

Our response: Thank you, I agreed that the interpretation may vary depending on the measures which is used to present the data. I have removed the statement of infectious diseases from the conclusion. Please read as “Our study indicates a variation of needs of emergency care by age groups and levels of care among patients admitted to CCUs in provincial and district hospitals in Vietnam”

8. There are no specific proposals or suggestions derived from the results of current study. Please add a paragraph in the discussion section about the application of the research finding in clinical practice and future research to compliment the limitations of current study.

Our response: Thank you, we appreciated this comment which is aligned with another reviewer’s comment. We have added the below text for the future research:

“Currently, there is limited cost-effectiveness analysis performed for critical care services in Vietnam. The results of this study in regards to the a variety of diagnoses and outcomes of admission will direct the future studies on the establishment of CCU registry, the effectiveness of CCU treatments and estimates of the total cost to save a life-year. This information will be helpful to develop policies and strategies to make better-informed decisions in resource-constrained settings in the face of an aging aged population in Vietnam.”

9. In the conclusion section, the authors mentioned that the medical needs varied by age group, but a detailed interpretation of the conclusion was not included in the discussion section. Fig 2 and 3 look fancy with color graphs, however, they only show the obvious and expected results of this study. Please add more detailed interpretations of the distribution of diagnoses and mortality by age groups in the discussion section.

Our response: Thank you, in the result section, we tried describing the distribution of diagnoses and outcomes by age groups and the conclusion came directly from the result. We recognized that this

information is quite obvious as you said then we would like to keep the discussion concise by only mentioning the current population age structure.

10. What is the meaning of supplementary Figure 1 (Pattern of admissions over months). It is not easy to follow. Please add the labels of X-axis and Y-axis and interpret this figure in discuss section.

Our response: The impacts of seasonal variation on the all causes, cardiovascular respiratory disease hospital admissions and mortality were noted in other studies (Sun et al. Thorax 2018;73:951-958; Lina et al. Int J Epidemiol. 2022 Feb 18;51(1):122-133). We provided the data to help readers understand the context of the study and may seres as a reference for future study. We also have added the labels for the clarity.

VERSION 2 – REVIEW

REVIEWER	Watters, David Deakin University, Surgery
REVIEW RETURNED	19-May-2022

GENERAL COMMENTS	I think this paper provides important information on critical illness, the pattern of disease and outcome in Vietnam. Although unfortunate that on 73% of patients had a hospital outcome recorded did your ICU's also collect data on ICU/Critical care outcome? If not it would be worth stating this was not available in the adminisitrative data you were accessing. It would be worth stating what information is available to you regarding the treatments received in the ICU to understand the survival of ventilated patients, patients requiring inotropic or renal support. It might also be worth knowing what proportion of patients in each GBD group required those levels of support.
--

REVIEWER	Kim, Won Young Asan medical center, emergency medicine
REVIEW RETURNED	05-May-2022

GENERAL COMMENTS	The authors response appropriately and revised quality of the article has increased. Thanks for your detailed response.
---

VERSION 2 – AUTHOR RESPONSE

Reviewer: 2

Dr. won young kim, Asan medical center

Comments to the Author:

The authors response appropriately and revised quality of the article has increased. Thanks for your detailed response.

Our response: We would like to thank the reviewer for their thoughtful and detailed comments which significantly improved the quality and readability of our manuscript.

Reviewer: 1

Dr. David Watters, Deakin University, Barwon Health

Comments to the Author:

I think this paper provides important information on critical illness, the pattern of disease and outcome in Vietnam. Although unfortunate that on 73% of patients had a hospital outcome recorded did your ICU's also collect data on ICU/Critical care outcome? If not it would be worth stating this was not available in the administrative data you were accessing.

It would be worth stating what information is available to you regarding the treatments received in the ICU to understand the survival of ventilated patients, patients requiring inotropic or renal support.

It might also be worth knowing what proportion of patients in each GBD group required those levels of support.

Our response: Thank you for these comments. We agree that it would have been desirable to add the ICU's outcome and interventions to help reader understand the context. Unfortunately, these information was not available. We have added a statement to declare available information as below in the method section.

"The data on the outcome at CCU discharge and interventions during CCU stay was not available for analysis."

VERSION 3 – REVIEW

REVIEWER	Watters, David Deakin University, Surgery
REVIEW RETURNED	26-May-2022

GENERAL COMMENTS	Thankyou for answering my question about CCU outcomes and the treatments recorded in your administrative data sets. I think this paper provides valuable information on the clinical conditions and their hospital outcomes being treated in Viet Nam CCU's. Congratulations on this work.
--